# Exposure to Plasticiser DEHP Affects Eggs Spawned by Blue Mussels: A Possible Risk to Fertilisation?

**DOI:** 10.3390/toxics12030172

**Published:** 2024-02-24

**Authors:** Luana Fiorella Mincarelli, Alexander Turner, George Anderson, Katharina Wollenberg Valero

**Affiliations:** 1Istituto Zooprofilattico Sperimentale dell’Abruzzo e del Molise “G. Caporale”, Via Campo Boario, 64100 Teramo, Italy; 2School of Natural Sciences, University of Hull, Hull HU6 7RX, UK; 3Department of Computer Science, University of Nottingham, Nottingham NG8 1BB, UK; 4School of Biology and Environmental Science, University College Dublin, D04 C1P1 Belfield, Ireland

**Keywords:** blue mussels, DEHP, fertility, reprotoxicity, egg count, egg size

## Abstract

The endocrine disruptive chemical DEHP is a plasticiser often found in marine waters. Here, we assessed the effect of this additive on the number and size of eggs spawned by female mussels during a synchronised spawning event. After achieving the ripeness of the gonads, mussels of both sexes were exposed to two environmentally relevant concentrations of DEHP (nominal concentrations 0.5 and 50 µg/L) for one week. A spawning event was then induced and eggs were collected, counted, and their size measured (area and diameter). A slight but not significant effect was observed in lowering the number of eggs spawned when increasing the DEHP concentration. This effect was greater when adding spent gonads (possibly fully spawned females) to the total number of females. A significant effect of the lower dose on the average egg sizes was noticed, with a smaller area and diameter measured with respect to the control and the higher concentrated treatments. These results once again underline the importance for ecotoxicological studies to address the nonlinear dose-response effects of endocrine disruptive chemicals environmentally present at concentrations in the order of just a few µg/L that could not elicit a strong defence mechanism at low levels and be absorbed by filter feeder animals such as mussels.

## 1. Introduction

Endocrine disrupting chemicals (EDCs) are exogenous substances or mixtures that usually alter the production and the function of hormones and receptors or interact with agonist or antagonist effects by hormone mimicking [1,2,3]. Generally, endocrine disruptive substances can be of both natural and anthropogenic in nature. Examples include oestrogens, phytoestrogens, plasticisers, pesticides, or herbicides [4,5,6]. Among the plastic additives, phthalates have been widely used as emollients and plasticisers in polyvinyl chloride (PVC) products [7,8,9], with an annual worldwide use of 8.4 million tonnes [10]. Phthalates could represent up to 50% of the total weight of certain plastic products [11,12] and are typically found in the environment, affecting habitats and organisms [13,14]. These additives leach from the plastic matrix as they are not chemically bonded to it, resulting in a consequent ubiquitous presence in the environment [7,15,16]. In aqueous environments, the migration of plastic chemical additives from everyday products has been highlighted by [17], who discovered several leachates in water samples that are able to cause oxidative response, antiandrogenic, and oestrogenic effects in humans and wildlife.

For humans, the skin route and inhalation are possible ways of phthalate absorption, but the food chain remains the most probable, in particular for phthalates with a long molecular chain such as di-2-ethylhexyl phthalate (DEHP, [16,18]). The daily human intake of DEHP has been measured to be in the range of 0.004–70 μg/kg/day [19], with detectable levels found in the blood, breast milk, umbilical cord, and urine [8]. This is relevant considering that plasticisers such as phthalates appear to shorten gestational duration, diminish sperm quality [20], and promote adipogenic activity, ultimately favouring obesity [21] in mammals. In [22], rats daily exposed by gavage to 9–48 mg/kg for 28 days displayed effects on the metabolic system and liver. Sex-related responses were also noted, with delayed reproductive development in males and effects on the thyroid for female rats. Oocyte apoptosis and a decrease in the number of primordial follicles were observed in newborn mouse ovaries cultured in vitro with 10–100 μM DEHP [23]. Fewer follicles were also counted in neonatal female mice intraperitoneally injected with 2.5–10 mg/kg/day [24]. Despite its restricted use in the European Union, especially in toys and childcare articles [25] and the evidence of its toxicity for aquatic [26,27,28,29] and terrestrial [24,30] species, DEHP still represents 40% of the global plastic softener market [10]. The natural production of DEHP by red algae (e.g., *Bangia atropurpurea*) was reported by [31], but the levels related to such natural sources are considered negligible compared to the massive levels of anthropogenic production [8]. In the past, traces of DEHP have also been detected in cosmetics and personal care products such as nail polishes, sanitary pads, fragrances, and baby lotions [32,33]. In natural conditions, the DEHP half-life persistence in surface water and sediments was evaluated to be 0.35–3.5 days [34]. Nevertheless, DEHP levels in Mediterranean coastal sites are often detected with published concentrations from 0.6 µg/L [35] to 71.6 µg/L [36], mainly due to the urban, industrial, and touristic impacts.

Molluscs such as mussels are considered non-controversial biomonitors and key species for their habitats as they are capable of the high bioaccumulation and bioconcentration of toxicants [37]. They are also common species for ecotoxicological experimental exposures to pollutants such as metals or anthropogenic chemicals [38]. Mussels are also used as distinctive indicators of health and food safety because of their position in the food chain and their close relationship with the human diet [39,40]. There are documented effects of plastic contamination on the reproductive cycle of *Mytilus*. As an example, styrene particles at concentrations of 0.01–1000 µg/L impaired the normal embryo development of *M. galloprovincialis* after a 48 h exposure [41]. Genotoxic effects of polystyrene microspheres at a concentration of 10^6^ particles/L were observed in the digestive glands and gills of *M. trossulus* after 5 days of exposure [42]. Furthermore, endocrine disruptor plasticiser BPA was noted to affect the full development of *M. galloprovincialis* larvae when the fertilised eggs were exposed to concentrations of 0.01–1000 μg/L [43,44].

Recently, published papers on the DEHP effect on molluscs have focused frequently on the consequences on the antioxidant system at various experimental timepoints, from 24 to 48 h in *M. galloprovincialis* [45] to 7–10 days in *Pinctada martensii* [46]. The DEHP effects on *Mytilus* spp. range from alterations in antioxidant and peroxisomal enzyme activities at high levels of 100–500 µg/L [47,48] to a hormetic effect on the expression of oestrogen receptor-like [49] and stress-related genes [50] when environmentally relevant concentrations are dosed. The hormetic effect is defined as a non-monotonic dose-response action of some endocrine-active chemicals such as DEHP, which display a stronger effect at low concentrations and a weaker or inhibited result at high levels [51,52,53]. The nonlinear dose responses of EDCs such as DEHP is one of the main challenges for regulatory agencies in the course of risk assessment. In fact, a linear extrapolation of the compound toxicity from high-dosed experiments is not valid in most cases, as the occupancy of receptor-mediated pathways can already saturate at low doses. Furthermore, xenoestrogens such as EDCs modulate a physiologically active system, which in most cases already acts above the threshold [54]. It is still unknown as to the range of environmentally relevant concentrations at which DEHP affects the reproductive characteristics in *Mytilus* spp. This paper aims to analyse the effect of short-term exposure to two environmentally relevant concentrations of DEHP on the number and size of eggs spawned from females during a synchronised reproductive event.

## 2. Materials and Methods

### 2.1. Experimental Design

Adult blue mussels *Mytilus edulis* (Linnaeus 1759, n = 90; length mean ± standard deviation = 5.7 cm ± 0.6 cm) were collected from the suspended rope farms of Cromarty Mussels Ltd. in Cromarty Firth, Scotland, UK (57.40.741 N 4.06.062 W) in April 2021 and transported to the aquarium facilities of the University of Hull. Mussels from the farm population were identified as *M. edulis* in [55], with molecular identification through PCR of the non-repetitive region *Mytilus* foot protein 1 *mfp-1* [56] to verify the presence of *M. edulis* (amplified segment at 180 bp). Neither hybrids with *M. galloprovincialis* (amplified segment at 126 bp) nor other *Mytilus* species were present. Mussels were neither cleaned from sand and mud nor scrubbed from seaweed and barnacles to avoid additional physical stress. Spring months were chosen for the experiment as this is the period with the highest likelihood of finding ripe gonads in mature to spawning stages [57]. In fact, in the Northern Hemisphere, the gametogenesis cycle of *M. edulis* starts in late autumn with declining temperatures, and the process continues over the winter months until maturation and spawning in spring, summer, or early autumn [58], depending on the geographic distribution [59], environmental conditions [60,61,62], nutrient availability, and energy reserves [63,64,65,66,67]. Thirty mussels for each of the three treatments were randomly divided into six 4-L continuously aerated glass tanks, for a total number of five mussels for each replicate tank at a density of one mussel per 0.8 L. Mussels were kept at laboratory conditions for 19 days in artificial saltwater (Premium REEF-Salt, Tropical Marine Centre©, Chorleywood, UK) in a final number of 18 continuously aerated glass tanks and fed with PhytoGreen-M phytoplankton (Brightwell Aquatics, Fort Payne, AL, USA). A longer acclimation period (compared to the other exposure experiments described in [49,55] was chosen in order to achieve ripeness of the gonads. Histological analysis to determine a baseline gametogenesis status and assess the potential ripeness of the gonads (and thus the readiness to the synchronised spawning event) was conducted on a subset of test mussels that were collected alongside the individuals used for this experiment and kept in an additional spare tank exposed to the same acclimation parameters. Mussels were kept in a climate-controlled room where the temperature was progressively raised by a total of 2 °C over the first week in order to avoid immediate temperature-induced shock and to facilitate the maturation of gonads, as was observed in [49], where an increase in temperature was noted to accelerate the mussel gametogenesis cycle.

After the acclimation period, mussels were exposed for seven days to two concentrations of DEHP (nominal concentrations 0.5 and 50 µg/L) for a final yield of three experimental treatments (CTRL (0 µg DEHP/L), LOW DEHP, HIGH DEHP). Exposures of 0.5 and 50 µg DEHP/L were chosen from the literature, in accordance with the levels found in coastal waters [35,36]. A 7-day DEHP exposure was chosen accounting for the non-persistence of DEHP in the environment [34,68]. Mussels were not fed during exposure, and artificial saltwater was prepared the day before each water change to allow the water temperature to adjust to the controlled room conditions. Water was partially (approx. 3 litres) changed every second day and DEHP was dosed right after (i.e., days 1, 3, and 5) from a stock solution of 1 mg/mL DEHP (≥99.5% purity, Sigma Aldrich^®^, Gillingham, UK) in ethanol (0.005% of the total water volume). Temperature, pH, and salinity were measured daily (Table 1) with a digital thermometer (Amarell Thermometer, Kreuzwertheim, Germany), a pH-metre (Jenway, Bibby Scientific Limited, Stone, UK), and a digital seawater refractometer (Hanna Instruments, Woonsocket, RI, USA). After seven days of exposure, the animals were stimulated to spawn as described below (unlicensed animal ethics approval reference no. #U080/FEC_2021_10, University of Hull). Eggs were collected from the tanks with a single-use pipette and mussel gonads were dissected. Approximately 1.0 cm^2^ of left gonad tissue was cut and immersed in 1 mL neutral-buffered 10% formalin solution (Sigma Aldrich, Gillingham, UK) at room temperature for histological observations.

### 2.2. Spawning Induction

During the fertilisation event, eggs are released as an intermittent pink cloud, while sperm is released in a thin, steady stream through the exhaling syphon cloud [69]. The KCl method from [70] was chosen as a spawning inductor, and carried out as follows: mussels were injected with 2 mL of 0.5 M KCl into the valve cavity and left for 2 h outside water. Then, they were randomly divided into 5 tanks for each treatment (n = 6 per replicate tank in a final volume of 2 L), considering that preliminary tests showed that mussels were more likely to spawn when placed with conspecifics and not in individual jars. Mussels were then left to spawn overnight in water.

### 2.3. Egg Counting and Size Measurement

Mussels were left to spawn overnight, and the morning after, water samples containing eggs were transferred into 2 mL tubes and fixed with formalin 10%. Eight aliquots of 10 μL each were immediately observed under a ZEISS Discovery Zen light microscope (Carl Zeiss, Cambridge, UK) with retro illumination and an Axiocam camera (Carl Zeiss, Cambridge, UK). Pictures of each aliquot (8 aliquots for each of the 5 replicate tanks for each treatment) were taken and later blindly analysed with the *ImageJ* software 1.8.0 (*cell_counter* plugin). Eggs were then counted and the size of one random egg per picture (i.e., 8 eggs for each of the 5 replicate tanks for each treatment) was measured using *ImageJ* 1.8.0. The egg diameter was then calculated from the area using the formula *diameter* (*D*) = (√*area* (*A*)/*Π*) × 2. Mussel individuals were then sampled and tissues for gonads were collected for histological analysis.

### 2.4. Histology Analysis to Determine Sex and Maturity Factor

Gonad samples from all individuals belonging to the experiment were fixed in 10% buffered formalin (Sigma-Aldrich, Gillingham, UK) and then washed with 0.01 M PBS (Sigma Aldrich, Irvine, UK), dehydrated with increasing ethanol (Fisher Scientific, Loughborough, UK) concentrations (70%, 90%, 100%) and cleared with Histoclear II (National Diagnostics, Atlanta, GA, USA). The day after, the samples were embedded in paraffin wax (VWR, Poole, UK) in an EG 1160 Paraffin Wax Embedding Centre (Leica Microsystems, Milton Keynes, UK) and tissue sections (10 µm) of wax-embedded gonads were cut on a Shandon Finesse^®^ Manual Rotary Microtome 325 (Thermo Fisher Scientific, Loughborough, UK). Slides were stained with Mayer’s haematoxylin solution (Sigma-Aldrich, Schnelldorf, Germany) and eosin Y alcoholic solution (Sigma-Aldrich, Schnelldorf, Germany). Prior to microscopic analysis, the microscope slides were coded in order to conduct a blind observation. Males and females were identified, and the following stages were blindly assessed following the stage descriptions reported by [57], where each stage was categorised by a maturity factor (MF): (i) spent/resting gonad (MF = 1, Figure 1G,H); (ii) development, stages 1 and 3; (iii) development stage 5 (mature/ripe gonads, MF = 3, Figure 1A,B); (iv) spawning stages 3 and 1 (MF = 4, Figure 1C–F). In particular, the following characteristics were recognised in females: mature/ripe gonads displayed compacted and polygonal-shape ova with some developing oocytes (Figure 1A), while spawning gonads showed empty spaces in the follicles, with some residual mature eggs still present (Figure 1 C,E). The sexual maturity index (SMI) was calculated according to the equation established by [71]: *SMI* = *Σ* (*proportion of each stage* × *maturity factor*). No particular alterations were found in the gonads between treatments after exposure to DEHP.

### 2.5. Respirometer Assay and Valve Behaviours

Behavioural and physiological analyses were carried out on adult blue mussels (n = 36; mean length 5.5 cm ± 0.45 cm) collected from the same population in Cromarty Mussels Ltd. (Cromarty Firth, UK) in January 2021 and kept for 19 days in the aquarium facilities of the University of Hull at T = 8.62 ± 0.28 °C, pH = 8.08 ± 0.03, and 35 ± 1 psu salinity. This additional analysis was performed to test the consequences on the behavioural and metabolic traits of blue mussels (i.e., respiration and valve movement) after a single-dose exposure to the plastic additive DEHP and whether a non-monotonic, nonlinear dose response could be related to the altered uptake or filtration activity of mussels. Mussels were tested on day 1 (CTRL), and the following day (day 2), the same groups were exposed to a single dose of either 0.5 μg/L (LOW DEHP) or 50 μg/L (HIGH DEHP) of DEHP as follows: (I) mussels were individually placed into 200-mL jars and acclimated for 5 min; (II) after acclimation, 10 μL of either control seawater (on day 1) or DEHP (0.5 or 50 μg/L on day 2) was injected in the proximity of the gills; (III) after lid closure, the valve behaviour was recorded on video, and oxygen consumption was measured with a USB-powered fibre optic metre FireStingO_2_ for a 5-min assay. Each mussel was tested once per day and at the same time over the two days in order to minimise animal stress and the possible effects of circadian rhythm on valve movements [72,73,74]. As an estimator for the basal metabolic rate, oxygen consumption (mg/L) was recorded by the respirometer every 30 s to yield a total of 10 data points over the 5-min assay for each individual. The 5-min interval was chosen considering the filtration rate of 46–80 mL/min (230–400 mL in 5 min) for *M. edulis* individuals of 5.65 cm in length [75]. The partial oxygen consumption Δmg/L for each time point t was calculated as follows: oxygen concentration (mg/L) at time point (t)-oxygen concentration (mg/L) at time point (t − 1). Before the experiment, blank tests were conducted in the same 200-mL jars in order to assess the background microbial water consumption from the experimental water tanks. During the same 5-min interval of the respirometer assay, the behaviours of the valves were recorded on video as behavioural changes are usually recorded at the same time as physiological assays as respiration is closely related to the activity and movements of the valves [76]. Videos were then coded, and behaviours were scored blindly in order to remove possible observer bias. The following valve behaviours were assessed for the same data points as the respirometer test over the 5-min assay (10 data points of 30 s each): (I) valve status: valves were either closed or open; (II) valve changes: number of changes between opening and closure or vice versa (none, one, more than one); (III) valve openings: number of only opening events (from closed to open, measure as none, one, more than one).

### 2.6. Statistical Analysis

Ordinal logistic regression was used to predict the ordinal dependent variables “Gametogenesis stage”, assuming “DEHP” and “SEX” as independent variables. The dependent variable “Gametogenesis stage” was measured at the ordinal level (i.e., a 3-point scale ranging from “development” to “mature” to “spawning”). Independent variables “DEHP” and “SEX” were considered categorical variables. Model uncertainty was assessed by comparing the ΔAICc values and Akaike weights, in which higher values for ΔAICc indicate second best to last parsimonious models of the set (Table 2). Model selection was carried out in Rstudio with the *AICcmodavg* package [77] in R 3.6.2 (CRAN). Models with ΔAIC > 10 were omitted from consideration since they have considerably less support compared to the best-fitting model [78]. Ordinal logistic regression was carried out using the *polr* function (*MASS* package, [79]), calculating the *p* value by comparing the *t*-value against the standard normal distribution (Table 3). The proportional odds assumption (test of parallel lines) was tested using the *ordinal* package [80]. Since the sex of mussels could not be determined through external morphology but only by histology, we investigated whether the number of eggs observed was correlated with the number of females in the tank post-histology. A significant correlation between the number of spawned eggs and the number of females in the tanks was found by the Pearson’s test (*p* < 0.001, t = 3.63). Therefore, the number of eggs per each of the eight aliquots was divided by the number of females present in the tank before undertaking the statistical analysis. One female in the HIGH DEHP treatment was taken out of the calculation due to displaying gonads in the early developing state, and thus unlikely to spawn when induced.

During the histological observations, some microscopic analysis of the follicles resulted in inconclusive results due to the display of undetermined sex traits (as shown in Figure 1G), characteristics of spent gonads. Considering that the observed undetermined gonads (n = 1 for CTRL, n = 1 for LOW DEHP, n = 4 for HIGH DEHP) could have been either spawned males or females and that a mussel population usually displays a 1:1 ratio between sexes, additional statistical analysis was carried out by adding 50% of the resting/spent gonads (i.e., n = 0.5 for CTRL, n = 0.5 for LOW DEHP, n = 2 for HIGH DEHP), considering them as possible fully spawned and spent females.

The relative number of eggs was then analysed via the nonparametric Kruskal–Wallis test (*stats* package) after verifying the non-normality (Shapiro–Wilk’s test) and homogeneity (Levene’s test) of the dataset. Dunn’s test, a post hoc test suitable for nonparametric data, was then used for comparisons between groups. Regarding the average egg size, the area and diameter were analysed with the ANOVA test (*stats* package) after verifying the normality (Shapiro–Wilk’s test) and homogeneity (Levene’s test) of the dataset. Tukey’s multiple comparison test, which is suitable for comparing the means of normally distributed data following parametric analysis, was then used for comparisons between groups.

Moreover, the partial oxygen consumption measurements over the 30-s data points were analysed by ANOVA for Randomised Block Design after data normalisation (*bestNormalize* package, [81]) to determine the impact of treatments while correcting for time points as a blocking factor. For valve behaviours, one-way PERMANOVA [82] using the Jaccard dissimilarity matrix and 9999 permutations (*vegan* package, [83]) was applied to the valve status (either “closed” or “open”), valve changes (number of changes between open and closed valves and vice versa), and valve openings (number of valve opening events) to test the effect of the DEHP exposures. All graphs were created using MATLAB R2022b.

## 3. Results

The most parsimonious ordered logistic regression model (SEX + DEHP) showed that there was a significant difference between sexes (*p* SEX = 0.002, *t*-value = −3.14) and in their SMIs. When analysing the effect of DEHP, there was no effect of the plasticiser on the transition between stages (*p* > 0.05, Figure 2, Table 2 and Table 3).

Similar to the sexual maturity index, the Kruskal–Wallis test highlighted a lowered but slightly not significant effect of DEHP on the number of spawned eggs by females (*p* = 0.10, KW chi-squared = 4.52, Figure 3).

The LOW and HIGH DEHP treatments showed the lowest average egg count in the 10 microlitre aliquots (57.1 ± 9.0 SEM eggs/female for LOW DEHP and 54.0 ± 8.9 SEM egg/female for HIGH DEHP), lower when compared to the control spawned eggs (77.0 ± 11.8 SEM eggs/female). Usually, *M. edulis* females emit ca. 10^6^–10^9^ eggs per female, depending on body size [84,85,86]. In this experiment, the female body sizes were coherent between treatments (5.6 ± 0.7 cm in CTRL; 5.7 ± 0.5 cm in LOW DEHP; 5.9 ± 0.6 cm in HIGH DEHP), with no significant difference between groups (one-way ANOVA *p* > 0.05). It is therefore unlikely that differing relative egg counts resulted from different body sizes.

When adding 50% of the undetermined gonads (i.e., possible fully spawned females, n = 0.5 for CTRL, n = 0.5 for LOW DEHP, n = 2 for HIGH DEHP) to the total number of females, the effect of DEHP was observed to intensify. In fact, adjusting the spawned eggs by the additional spent gonads resulted in a mean ± standard error of the mean SEM of 75.8 ± 12.0 eggs/female in CTRL, 52.8 ± 8.1 eggs/female for LOW DEHP, and 33.3 ± 4.9 egg/female for HIGH DEHP. In this case, the Kruskal–Wallis test uncovered a significant effect of DEHP in lowering the egg count (*p* = 0.02, KW-H= 8.02, Figure 4).

For the egg area measurement (squared micrometre), ANOVA found an effect of DEHP on the egg area (*p* < 0.001, F value = 7.63, Figure 5 and Figure 6). Tukey’s test highlighted a significant difference between the control eggs and those exposed to a low concentration of DEHP (*p* = 0.04) and between the low and high DEHP treatment groups (*p* < 0.001). In fact, LOW DEHP exposure had a significant effect in lowering the size of the eggs, with the smallest cells observable in the tanks exposed to the low concentration of DEHP (area = 2853 ± 67 SEM μm^2^; diameter = 60.1 ± 0.7 SEM μm), while the high concentration treatments were of similar size to the control condition tanks. Specifically, the average values for the HIGH DEHP groups were 3377 ± 100 SEM μm^2^ for the egg area and 65.3 ± 1.0 SEM μm for the egg diameter, while in the CTRL condition, the average area was 3200 ± 101 SEM μm^2^ and the diameter = 63.5 ± 1.0 SEM μm.

Regarding the respirometer assay, the HIGH DEHP dose was found to be significantly effective in increasing the oxygen consumption with respect to the CTRL, especially in the first 180 s after injection in the proximity of the gills (*p* < 0.001, F value = 13.53). Even though not significant, a lowered oxygen consumption was also observable in the LOW DEHP mussels with respect to the CTRL group, especially in the first 180 s, highlighting an inverted trend of the two concentrations on mussel oxygen consumption (Figure 7).

No significant effects were observed for the low and high DEHP dose for valve movements (*p* > 0.05, Figure 8). A trend in valves being closed more often was noted for the LOW DEHP-treated mussels against the CTRL, both, however, were not statistically significant.

## 4. Discussion

Exogenous factors such as temperature, food availability, nutrient quality, salinity, circadian rhythm, or tides could control certain aspects of the reproductive cycle such as duration and periodicity of the gametogenesis and larval stages [87,88,89,90]. Endogenous factors and species-specific characteristics such as animal hormone levels or individual responses to environmental conditions might also affect and regulate gametogenesis or spawning [91]. Moreover, gametes in the water represent a chemical stimulus to ripe mussels in order to induce spawning and increase the success of fertilisation [58]. As shown in [49,55], DEHP in environmentally relevant concentrations does not seem to induce any alteration of the gametogenesis stage in males or females. Likewise, here, even though a small decrease in the SMI for females was noted with exposure to increasing concentrations of DEHP, it was not significant. This does not preclude additional dysfunctions from the endocrine disruptor DEHP on reproductive traits, as already shown for fish [26,92] or crustaceans [93,94] at various concentrations from 0.02 to 500 µg/L.

A lowered but slightly not significant effect of DEHP was noted on the number of spawned eggs. This attenuated response could be related to the fact that DEHP was administered only during the terminal part of the gametogenesis cycle, when females were already in the final developing phase and the gametes were already present in the gonads and just needed to grow to maturation, hence there was only very little effect on the number spawned. However, it is important to highlight that when adding spent gonads as possible fully spawned females to the statistical analysis, the effect of DEHP in lowering the number of counted eggs was further pronounced, displaying a significant difference between the treated individuals with respect to the control. In line with these results, when exposed for a year to concentrations of 10–100 ng/L of tributyltin (TBT), adult periwinkle *Littorina littorea* showed decreased egg production, but with marginal effect after a short-term exposure of 9 days [95]. The exposure to the antiandrogenic compound flutamide over a period of 21 days resulted in a decrease in eggs spawned per female due to a delay in the maturation of eggs in fathead minnow *Pimephales promelas* [96]. Significant reductions in eggs spawned for females were also noted in *Danio rerio* exposed long-term to concentrations >1.67 ng/L and 1500 μg/L of 17α-ethinylestradiol (EE2) and BPA, respectively. The effects were often associated with increased vitellogenin plasma levels and gonadal alterations, while exposure to the same compounds for a shorter time of 0–3 days post-fertilisation provoked no effect on spawned eggs [97]. A similar reduction in female fecundity was found for *D. rerio* exposed to nonylphenol (100 μg/L) or EE2 (10 μg/L) for two months [98]. These effects could be caused by interference with mitosis, cell cycle progression, protein metabolism, and/or the final maturation of oocytes [99]. The results from the literature might suggest a possible effect of EDCs on the number of spawned egg if the females are exposed for a prolonged period over their gametogenesis, even though a 7-day exposure was noted here to already induce a slight decrease in the DEHP-treated groups.

For the egg area measurement (squared micrometre), there was an effect of DEHP on lowering the egg size. This seems to confirm the non-monotonic dose-response effect of endocrine disruptive chemicals such as DEHP. In the literature, the egg diameter of *Mytilus* eggs is reported as being around 70 μm [84], and more specifically, fertilised eggs are usually 60–65 μm in diameter [58]. With respect to the CTRL and HIGH DEHP eggs, a higher number of eggs shown for the LOW DEHP treatment fell below the threshold of 60 μm. This could indicate a lower fertilisation rate with respect to the other two conditions, but further studies are needed to address this particular risk to mussel reproductive outcomes. Chemicals such as plasticisers or synthetic drugs are defined as selective modulators of the endocrine system. This means that they often cause nonlinear responses and either interfere with the synthesis and/or metabolism of hormones and their receptors [100,101]. In some cases, the resulting dose responses follow a biphasic curve characterised by stimulation at low doses and inhibition at higher doses [102]. In fact, it is well-known that some pollutants present dose responses that show not only linear, power, or exponential distributions, but also a U-shape, inverted U-shape, J-shape, and inverted J-shape [102]. Inverted U-shape dose–response curves were, for example, observed for cadmium exposure on the activity of superoxide dismutase (SOD) and catalase (CAT) in the earthworm *Eisenia fetida*, while higher doses provoked the inhibition of these antioxidant enzymes, possibly related to the activation of pathways of adaptation [103]. Several studies have reported that low-concentration exposure to xenobiotics such as heavy metals could elicit an adaptive mechanism characterised by increasing energy storage, which is overcompensated by excessive energy consumption at higher doses, to balance the energy metabolism and maintain homeostasis [104,105,106,107]. In [108], the sperm and embryos of sea urchins *Paracentrotus lividus* and *Sphaerechinus granularis* exposed to vegetal- and chemical-based tannins resulted in a general initial increase in the fertilisation rate at concentrations of 0.1–0.3 mg/L and a shift to toxicity at higher concentration doses. These types of hormetic effects are also induced by natural and environmental factors such as temperature, ground-level ozone, magnetic field, and radiation [109]. It is interesting to note that when increasing the exposure concentrations, DEHP elicited two different response curves: a linear one for the number of eggs spawned and a biphasic curve for the egg sizes. Considering the number of spawned eggs, it is important to also note a slight trend towards a lower proportion of spawning female gonads with high DEHP concentration, which could also have affected the egg number.

In bivalves, gonads consist of branching tubules united to form ducts (that eventually lead into a short gonoduct), with gametes situated in the epithelial lining that are subsequently shed into the water through the exhaling mantle opening [58]. In females, primary oogonia follow repeated mitosis to secondary oogonia (5–7 μm diameter), which as primary oocytes undergo meiosis until it is arrested at prophase I (as the remaining part of meiosis is completed at fertilisation). Then, vitellogenesis takes place, when oocytes accumulate nutritive substances such as lipid globules, vitellogenin (egg yolk vitellin precursor), and cortical granules [110]. In invertebrates with internal fertilisation such as *Caenorhabditis elegans* nematodes, it was recently described that low concentrations of DEHP can alter meiotic processes such as increased meiotic double-strand breaks, defects in chromosome remodelling at late prophase I, aberrant chromosome morphology in diakinesis oocytes, and increased chromosome non-disjunction [111]. Contrary to our results, in [112], exposure to the oestrogenic compound E2 of the Pacific oyster *Crassostrea gigas* for 40 days increased the diameter and vitellin content in the oocytes, suggesting that the chemical E2 is involved in the vitellogenesis in female oysters. This could suggest that DEHP in mussels might negatively affect the same pathway, affect the oocyte reserves, and eventually, the size of the spawned eggs. Considering this, it is possible that in this experiment, low DEHP exposure for one week affected the female oocytes by disrupting the meiotic process and/or the accumulation of nutritive deposits during the final phase of vitellogenesis, eventually impacting the egg size.

Exposure to external chemicals could also affect the valve activity and respiration rate, eventually leading to changes in the filtration and absorption of toxicants. Here, the HIGH DEHP dose was able to induce an increase in the respiration of mussels, especially in the first minutes of exposure. On the contrary, this effect was not noticed for the LOW DEHP exposure. Increased respiration rate was also observed in Wang et al. [113] when mussels were exposed to plastic particles. This was explained as enhanced metabolic compensation in response to environmental stressors, suggesting that a higher concentration of the plasticiser DEHP could be sensed as a more threatening stimulus by the mussels compared to the LOW DEHP dose, which could, in turn, bypass the defence mechanisms and be absorbed by the animals during filtration, thus eliciting a stronger response in reproductive outcomes.

## 5. Conclusions

In conclusion, a non-monotonic dose-response was observable for the plastic additive DEHP during the synchronised reproductive event, especially on the size of the eggs spawned. Females exposed to a low concentration of DEHP (nominal concentration 0.5 µg/L) for seven days spawned smaller eggs compared to the control and the higher concentration (nominal concentration 50 µg/L, with regard to the measured egg area) groups. Regarding the egg count, no significant effect was noted, even though the number of spawned eggs decreased in the DEHP-treated groups, with the effect further pronounced when considering undetermined sex gonads as possible fully spawned females. As already observed, a linear toxicity response was not observable for endocrine disrupting chemicals such as DEHP, highlighting the importance of ecotoxicological studies to address the effect of EDCs at low concentrations, which are the most prominent levels found in natural environments. In fact, as this paper and many other studies before have underlined, DEHP and other endocrine disruptors (used in herbicides, fungicides, insecticides, UV screens, lubricants, and paints) could stimulate stronger effects and affect organisms at low environmental concentrations. It is already known that non-monotonic responses to endocrine-active chemicals are often related to receptor actions or the presence of antagonists, regulators, and co-activators [55,109,110]. From these preliminary results, it could be also hypothesised that low concentrations exhibit strong responses as they are better absorbed by filter feeders such as mussels, as higher levels may more likely be sensed as a threat and rapidly excreted or cause metabolic or behavioural changes. However, what we did not investigate in this study was whether there are altered baseline tolerances of these populations based on natural occurrences of environmental toxins. Thus, policies regulating the use and disposition of these chemicals, along with the setting of new safe environmental levels, should be developed.

## Figures and Tables

**Figure 1 toxics-12-00172-f001:**
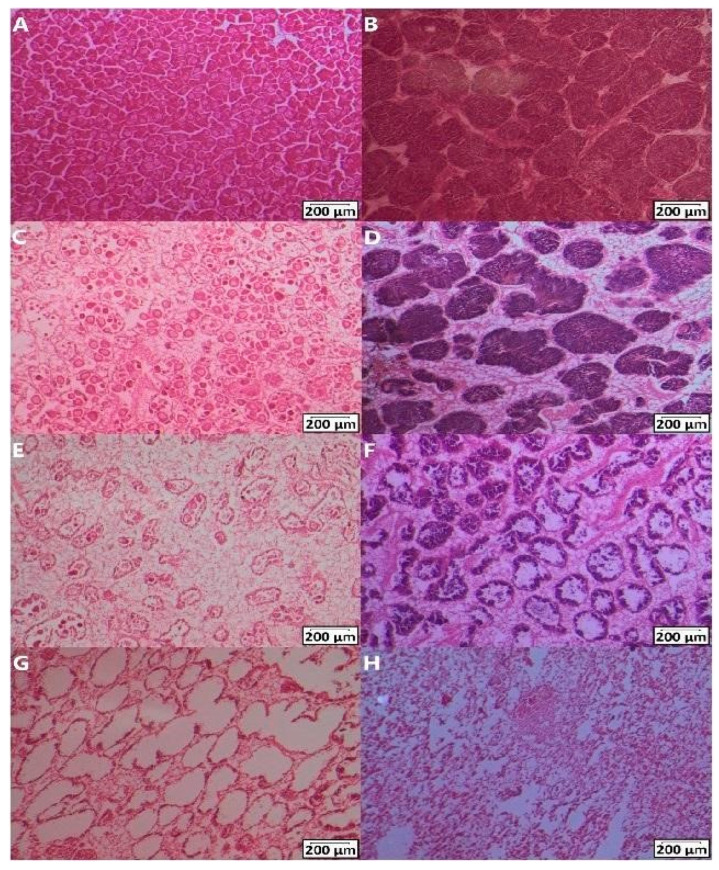
Gametogenesis stages of 10 μm gonadal tissue sections after DEHP exposure and the spawning induction in males and females stained with haematoxylin and eosin. Mature gonads in females (**A**) and males (**B**), spawning stage 3 in females (**C**) and males (**D**), spawning stage 1 in females (**E**) and males (**F**), empty spent follicles (**G**), and resting/spent stage (**H**). Scale bars represent 200 μm. Images were modified for brightness and contrast.

**Figure 2 toxics-12-00172-f002:**
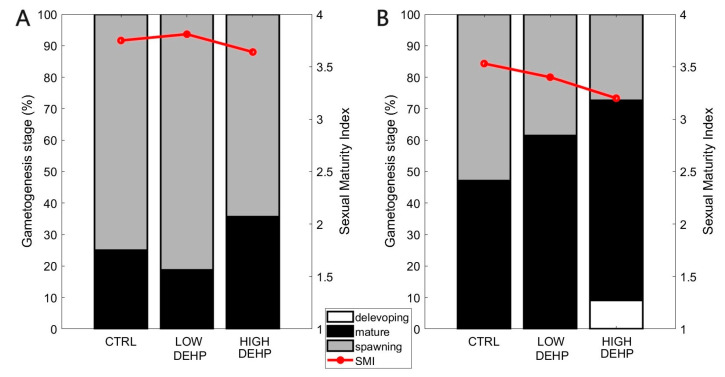
Effects of DEHP treatments on the gametogenesis stages. Percentage of each stage and sexual maturity index (SMI) of males (**A**) and females (**B**) in CTRL (n = 13 (males), 15 (females)) LOW DEHP (n = 17 (males), 10 (females)), HIGH DEHP (n = 9 (males), 14 (females)), and the associated SMIs.

**Figure 3 toxics-12-00172-f003:**
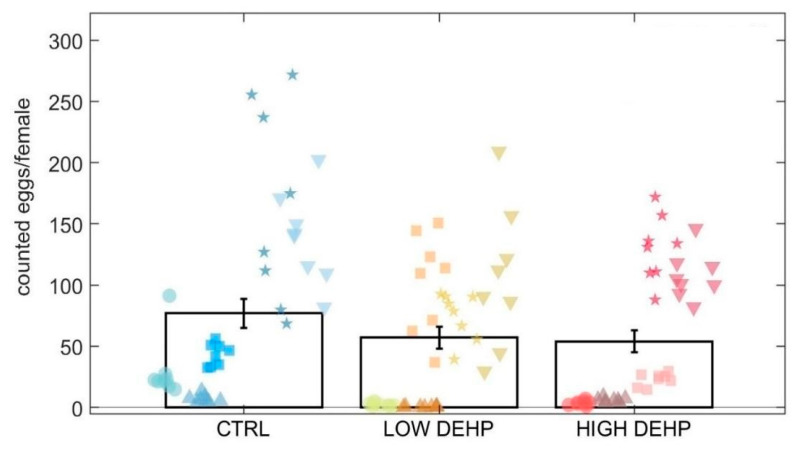
Total eggs for females counted in 10 microlitres (8 replicate aliquots counted in 5 tanks for each treatment). Data are expressed as the mean ± standard error of the mean (SEM). Abbreviations are CTRL (0 µg/L), LOW DEHP (0.5 µg/L), and HIGH DEHP (50 µg/L). Different shapes represent different replicate tanks.

**Figure 4 toxics-12-00172-f004:**
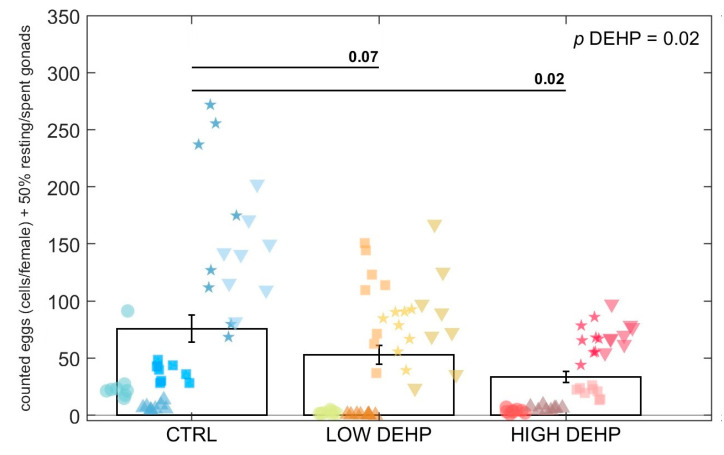
Counted eggs adjusted by adding 50% of the observed spent gonads to the total females in 10 microlitres (8 replicate aliquots counted in 5 tanks for each treatment). Data are expressed as the mean ± standard error of the mean (SEM). Abbreviations are CTRL (0 µg/L), LOW DEHP (0.5 µg/L), and HIGH DEHP (50 µg/L). Different shapes represent different replicate tanks. Kruskal–Wallis *p* value is annotated in the top right corner.

**Figure 5 toxics-12-00172-f005:**
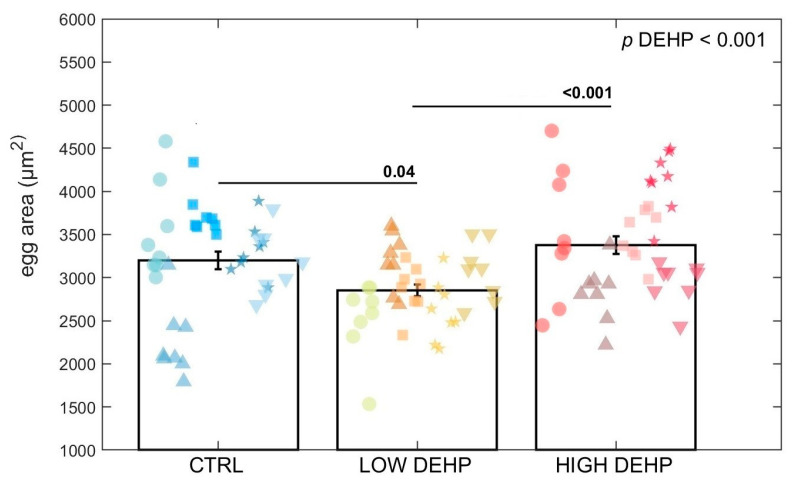
Egg area (squared micrometre) for each treatment (8 replicate aliquots measured in 5 tanks for each treatment). Data are expressed as the mean ± standard error of the mean (SEM). Abbreviations are CTRL (0 µg/L), LOW DEHP (0.5 µg/L), and HIGH DEHP (50 µg/L). Different shapes represent different replicate tanks. ANOVA *p* values are annotated and differences between groups are indicated by bars over the histograms.

**Figure 6 toxics-12-00172-f006:**
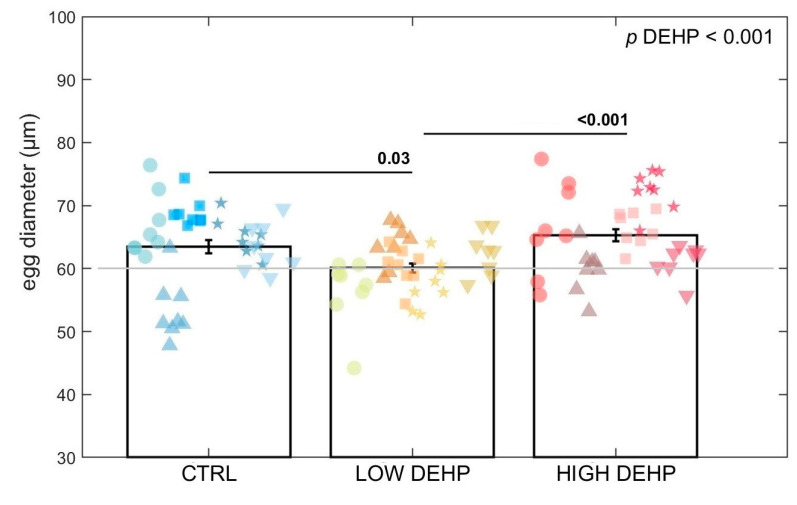
Egg diameter (micrometre) for each treatment (8 replicate aliquots measured in 5 tanks for each treatment). Data are expressed as the mean ± standard error of the mean (SEM). Abbreviations are CTRL (0 µg/L), LOW DEHP (0.5 µg/L), and HIGH DEHP (50 µg/L). Different shapes represent different replicate tanks. The grey line represents the minimum threshold of fertilised eggs reported by [58]. ANOVA *p* values are annotated and differences between groups are indicated by bars over the histograms.

**Figure 7 toxics-12-00172-f007:**
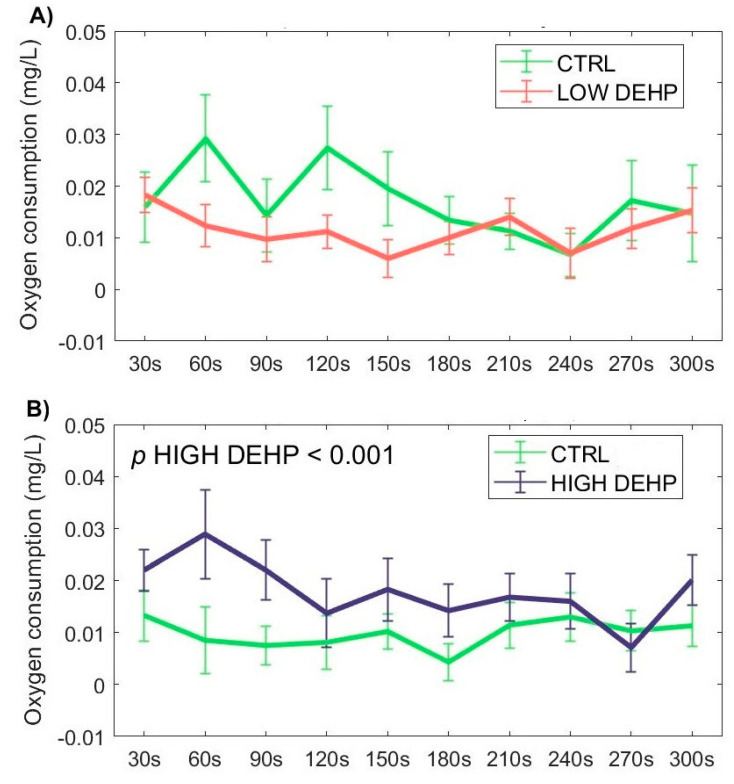
DEHP effect on the mg/L water oxygen concentration (consumed oxygen estimation) throughout the 30−s time points on day 2 after an injection of (**A**) 0.5 µg DEHP/L or (**B**) 50 µg DEHP/L.Abbreviations are CTRL (0 µg/L, in green), LOW DEHP (0.5 µg/L, in red), and HIGH DEHP (50 µg/L, in blue), n = 17−18. Datapoints are expressed as the mean ± standard error of the mean (SEM). ANOVA significant *p* values are annotated on the graphs.

**Figure 8 toxics-12-00172-f008:**
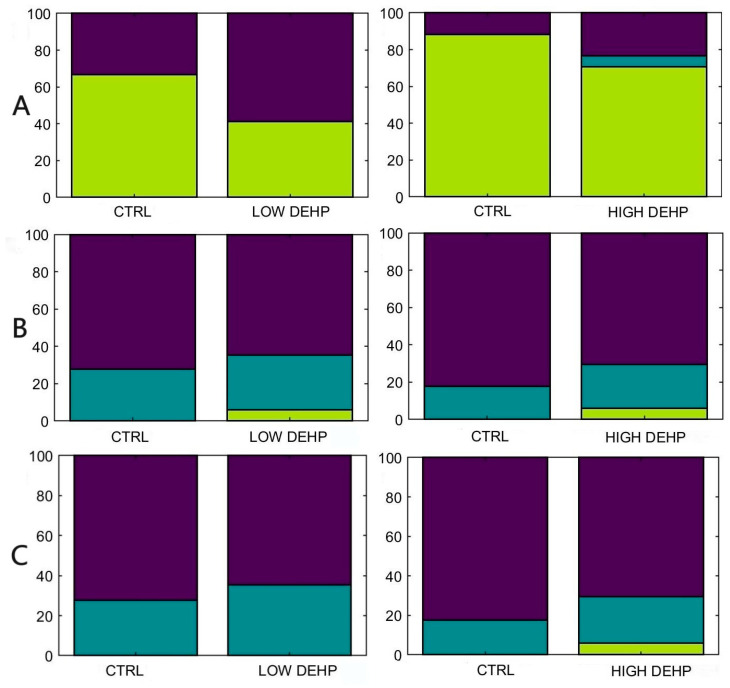
Percentages of individuals with (**A**) open (green) or closed (purple) valves (modal values). “Split” (blue) values indicate mussels that have spent an equal time in open and closed states exposed to each of the two DEHP concentrations. (**B**) Percentages of individuals with more than one change from open to closed valves or vice versa (green), one change (blue) or no changes from the initial status (purple). (**C**) Percentages of individuals with more than opening events (from closed to open), measured as none (purple), one (blue), more than one (green). Abbreviations are CTRL (0 µg/L), LOW DEHP (0.5 µg/L), and HIGH DEHP (50 µg/L), n = 17–18.

**Table 1 toxics-12-00172-t001:** Experimental treatments and measurements of temperature and pH at salinity values of 35 ± 1. All parameters are expressed as mean ± standard deviation.

Name of Treatment	Temperature (°C)	pH (Units)
CTRL (0 µg DEHP/L)	12.8 ± 0.4	7.9 ± 0.1
LOW DEHP (0.5 µg DEHP/L)	12.6 ± 0.3	7.8 ± 0.2
High DEHP (50 µg DEHP/L)	12.7 ± 0.3	7.8 ± 0.1

**Table 2 toxics-12-00172-t002:** Model classification, number of estimated parameters (K) for each model, Akaike information criterion (AICc), delta AIC (ΔAIC), Akaike weights (AICcWT), cumulative Akaike weights (CumWT), and log-likelihood of each model (LL) for the two independent variables (+) sex (SEX) and DEHP concentration (DEHP) and their interactions (*) on the gametogenesis stages.

Model (K)	AICc; ΔAIC; AICcWT	Cum WT; LL
SEX + DEHP (4)	118.15; 0.00; 0.47	0.47; −54.99
SEX (3)	118.82; 0.67; 0.67	0.80; −56.35
SEX * DEHP (5)	119.89; 1.74; 0.20	0.99; −54.82
DEHP (3)	126.92; 8.78; 0.01	1.0; −60.41

**Table 3 toxics-12-00172-t003:** Results of the ordinal logistic regression for the best model of treatments (SEX). Estimated value, standard error, *t*-value, and *p* value for the independent variables sex and DEHP concentration (DEHP).

Variable	Value; Std. Error	*t*-Value; *p* Value
SEX	−1.56; 0.50	−3.14; 0.002
DEHP	−0.50; 0.31	−1.62; 0.104

## Data Availability

Original data and materials from this paper could be available upon reasonable request.

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
