# Peer review of "Exposure to Plasticiser DEHP Affects Eggs Spawned by Blue Mussels: A Possible Risk to Fertilisation?"

_toxics, 2024, doi:10.3390/toxics12030172_

Round 1

Reviewer 1 Report

Comments and Suggestions for Authors

The paper is well-written and provides sufficient background and importance for conducting the study. Further characterization of the interpretation behind the non-monotonic response(s) seen for some measures could help to strengthen its overall purpose. There is a small concern in the statistical power for the egg number analysis, given the inability to distinguish and equally randomize males and females across exposure and spawning aquaria. There are also some questions as to the interpretation of the egg size data. Itemized comments are listed below.

Introduction

P2, L48-53: Are there additional references/information that could be provided that link the described reproductive and developmental effects with a given concentration of DEHP, for comparison with the daily human intake? If so, this would add to the relevance of the concentrations tested in this given study and those found naturally in the environment.

P2, L62-66: The environmental half-life of DEHP seems to be short. Is there additional information or interpretation that could be provided regarding the apparent discrepancy between this low persistence and high concentrations in the referenced studies? For example, coastal areas should be more prone to impact from anthropogenic sources than more distant marine environments, but this statement suggests the opposite.

Methods

P3, L125-126: Are there data on concentrations of DEHP or other exogenous chemicals at the site of sampling? Would it be possible that if such chemicals are present, mussels would be acclimated to such conditions and present a less robust response within the study conditions? Since the site is listed as a “farm”, it is assumed that it would be near pristine, but such data would add to the site description.

P3, L129-131: Were the administered concentrations tested analytically? With such low persistence, even with water changes and chemical addition every 2 days, it may be possible for large reductions in chemical concentrations over time or for binding to glass surfaces of the aquaria.

General comment: As the sex of mussels could not be determined until histopathological evaluation, is there concern that even with random allocation to exposure aquaria and spawning aquaria that 1) variation among tank concentrations could impact sexes unequally (i.e., higher concentration in tank with predominantly females); or 2) that statistical power is reduced with non-equal numbers of males and females per spawning tank even when normalizing number of eggs by number of females per tank and use of non-parametric analyses?

General comment: It is assumed that histopathological evaluations were not conducted on a subset of individuals to determine baseline gametogenesis stage or sexual maturity index, based on the methods provided. Ideally, the control animals would account for any changes relative to the treated animals. However, it would still be useful to establish baseline measurements for these parameters to compare against test results of the study.

Results

P8, L251-252: The reasoning behind addition of undetermined gonads to the statistical analyses remains unclear. If the high DEHP females with undetermined gonads are 4X greater in number than from the control, then adding females not expected to produce eggs/spent seems to artificially reduce the ratio of #eggs/female (i.e., 693 eggs from 9 females for control and 486 eggs from 9 females for high DEHP when not adding undetermined gonads vs. 693 eggs from 10 females and 486 eggs from 13 females with the addition).   

Figure 3 and 4: Related to the statistical power comment above, it appears that there is higher variability within two replicate tanks for the control (range of ~55-275 eggs/female) driving the trend being seen for number of eggs. In most cases, the other two to three replicate tanks are grouped tightly together across the different treatments, and the majority of data points are within or slightly above their respective histograms (i.e., below or near the mean). With the differences in variability across the treatments, even with the non-parametric analyses, it is difficult to discern differences across the treatments. The results for the egg area and diameter are likely more reliable, given the similar variability seen across all treatments.

P8, L267 and Figure 5: There is a discrepancy between the two p-values listed for the low DEHP group; the text states 0.004, while the graph shows 0.04. 

Discussion

P10, L304-306: Related to comment above regarding baseline histopathology evaluation, such an evaluation could support the assumption of gametes already present in gonads at the time of exposure, especially given evidence of little influence of DEHP on gametogenesis.

P10, L334-335: While the eggs in the low DEHP treatment were smaller, they still seemed to be in the range provided in the citation (albeit at the low end), so how appropriate is it to assume lower fertilization? Could the smaller size just be natural variability within that given range, or what impact might female mussel size or gonad maturation state have on egg size? Providing this data or determining a correlation with egg size would help to support this assumption.

P11, L357-363: It is unclear how a U-shaped curve for egg size (an endocrine related outcome) is related to U-shape curve for antioxidant gene expression (a general response to xenobiotic insult) in a different species. This comparison could be characterized further (e.g., oxidative stress leading to lower maternal fitness?), especially regarding the non-monotonic response for both.

P11, L379-381: Considering that embryogenesis is a post-fertilization process, it would be helpful to further expand on how DEHP impacts this in nematodes, especially since it is stated further above that the smaller egg size in the low DEHP treatment may be due to a lower fertilization rate. Alternatively, the item regarding embryogenesis could be removed, as it does not relate to the next sentence on L381-384.

Conclusions

P11, L386-390: It is incorrect to state that fewer eggs were spawned in the low DEHP treatment relative to the high treatment and that there is a non-monotonic response to that measure. Similar, if not more eggs were spawned in the low treatment.

P12, L397-400: As this statement is the primary point of the article, it should be further expanded on to characterize what is driving these stronger responses at lower concentrations. It is briefly touched upon in the introduction, such as receptor saturation, but additional language here could significant strengthen the article and emphasize the importance of considering toxicity to marine bivalves at environmentally relevant concentrations. 

Author Response

Reviewer n. 1

Comments and Suggestions for Authors

The paper is well-written and provides sufficient background and importance for conducting the study. Further characterization of the interpretation behind the non-monotonic response(s) seen for some measures could help to strengthen its overall purpose. There is a small concern in the statistical power for the egg number analysis, given the inability to distinguish and equally randomize males and females across exposure and spawning aquaria. There are also some questions as to the interpretation of the egg size data. Itemized comments are listed below.

Response: We thank the reviewer for this thorough review of our manuscript. We appreciate the time and effort invested in this feedback. We now have provided further clarifications to the comments in the following point-by-point response.

Reviewer 2 Report

Comments and Suggestions for Authors

The manuscript toxics-2844568 submitted by Mincarelli et al. entitled "Plasticiser DEHP affects fertility and reproductive outcomes in female blue mussels submitted by Mincarelli et al. represents excellent article worthy of investigations and within the scope of MDPI Toxics Journal. The aim of this work is to analyse the effects of short-term exposure to two environmentally relevant concentrations of the plasticizer DEHP on the number and size of eggs spawned by female blue mussels during a synchronised reproductive event. In general, the effects of EDCs substances of natural or anthropogenic origin on wild species and especially on humans are a hot topic but receive less attention than necessary. Major comments Please prove and discuss that the observed non-monotonic, non-linear dose-response of DEHP is not due to the lower uptake and filtration activity of mussels in glass containers with high DEHP concentration Did you observe a difference in filtration activity between the control and low and high DEHP mussels? Minor Line 101 ->”Adult blue mussels Mytilus edulis Linnaeus, 1759 (N 90;” Provide a literature reference that the mussels collected were previously identified as M. edulis. Do the sibling species of the M. edulis complex (M. edulis, M. galloprovincialis and M. trossulus) and their hybrids possibly react differently to EDCs substances? Line 121 ->”to two concentrations of DEHP” Figures 3, 4, 5, 6 In the description of these figures, instead of “low DEHP (LOW DEHP) high DEHP (HIGH DEHP)” write “low DEHP (0.5 μg/L ) high DEHP (50 μg/L)”

Author Response

Reviever n. 2

Comments and Suggestions for Authors

The manuscript toxics-2844568 submitted by Mincarelli et al. entitled "Plasticiser DEHP affects fertility and reproductive outcomes in female blue mussels “ submitted by Mincarelli et al. represents excellent article worthy of investigations and within the scope of MDPI Toxics Journal. The aim of this work is to analyse the effects of short-term exposure to two environmentally relevant concentrations of the plasticizer DEHP on the number and size of eggs spawned by female blue mussels during a synchronised reproductive event. In general, the effects of EDCs substances of natural or anthropogenic origin on wild species and especially on humans are a hot topic but receive less attention than necessary. 

The authors thank the reviewer for the interesting review observations and for the kind evaluation that helped improve the manuscript. Below, we are giving a point-by-point itemised response to each of the reviewer’s comments with page and line number references. 

Reviewer 3 Report

Comments and Suggestions for Authors

DEHP is a common endocrine disruptor, and this manuscript investigates its impact on the fertility of blue mussels, which holds significant theoretical importance and is suitable for this journal. However, the manuscript has some weaknesses in the experimental design, methodology, and presentation that need further improvement.

1.     Based on the research content, the title of the paper needs further focus.

2.     Abstract: The presentation of the experimental methods and results is incomplete.

3.     Introduction: The existing review of the research background on the effects of DEHP on mollusks is fewer.

4.     Materials and Methods: Was the actual concentration of DEHP in the water measured? If so, what method was used, and what were the results?

5.     Tables 1, 2, and 3 should be presented in a three-line format.

6.     At which stage of gonad development were the blue mussels used in the experiments? Is there synchrony in gonad development among individuals? If there is asynchrony, comparing the results of each group is meaningless.

7.     Statistical Analysis: The author used Dunn's and Tukey multiple comparison tests in this manuscript. Could you clarify which type of multiple analysis results are presented in the manuscript?

8.     Were gonad samples collected from all blue mussels in each group? Since the author analyzed the developmental stage of each individual's gonad, what were the results regarding the gonad histology morphology? Can DEHP exposure also affect gonad histology morphology?

9. The author primarily investigates the effects of DEHP on the fertility of blue  mussels. Therefore, the discussion section should also incorporate relevant research for a comprehensive discussion. The inclusion of unrelated research indicators, such as changes in antioxidant enzyme activity in aquatic animals under heavy metal stress, is not meaningful.

Author Response

Reviewer n.3

Comments and Suggestions for Authors

DEHP is a common endocrine disruptor, and this manuscript investigates its impact on the fertility of blue mussels, which holds significant theoretical importance and is suitable for this journal. However, the manuscript has some weaknesses in the experimental design, methodology, and presentation that need further improvement.

Response: We thank the reviewer for the comments and suggestions, which we have now taken into account and have modified the manuscript accordingly.

Round 2

Reviewer 1 Report

Comments and Suggestions for Authors

The paper is much improved, especially with the addition of the respirometer tests and the consideration of metabolic changes/valve closures resulting in the non-monotonic response observed. In freshwater mussels it has been seen that high doses result in valve closure and protection from the toxicant at the expense of oxidative stress, until such a threshold is reached that the mussels must then open their valves again. At lower doses, detection may be limited, leading to longer periods of open values and higher exposure to the toxicant. Females of lure-displaying species are especially susceptible as they must retain open valves to attract fish hosts. It is interesting to see in this paper how marine mussels react. 

Figure 1: Minor typo, but 1E should say spawning "stage" 1.

Author Response

The authors thank again the reviewer for the interesting observations and for the kind evaluation that helped improve the manuscript. The typo in Figure 1 was corrected.

Reviewer 3 Report

Comments and Suggestions for Authors

Authors addressed most of the necessary corrections related to my specific comments and thoughts in the revised manuscript. However, there are still some issues that need to be addressed. Such as, the gonadal morphology structure shown in figure 1 represents the gonadal morphology of animals before the stress test or the gonad morphology after the stress test; the author should specify. I would agree to accept the revised manuscript after minor revision.

Author Response

We appreciate the time and effort invested in this feedback. Figure 1 was amended as follows:

Figure 1: "Gametogenesis stages of 10 μm gonadal tissue sections after the DEHP exposure and the spawning induction in males and females stained with haematoxylin and eosin. "